# Interest of Integrated Whole-Body PET/MR Imaging in Gastroenteropancreatic Neuroendocrine Neoplasms: A Retro-Prospective Study

**DOI:** 10.3390/cancers16132372

**Published:** 2024-06-28

**Authors:** Camelia Abid, Jenny Tannoury, Mathieu Uzzan, Edouard Reizine, Sébastien Mulé, Julia Chalaye, Alain Luciani, Emmanuel Itti, Iradj Sobhani

**Affiliations:** 1Department of Gastroenterology, Henri Mondor Hospital, 1 Rue Gustave Eiffel, 94000 Creteil, France; cameliaahlem.abid@gmail.com (C.A.); jenny.tannoury@gmail.com (J.T.); mathieu.uzzan@aphp.fr (M.U.); 2Department of Radiology, Henri Mondor Hospital, 94000 Creteil, France; edouard.reizine@aphp.fr (E.R.); sebastien.mule@aphp.fr (S.M.); alain.luciani@aphp.fr (A.L.); 3Department of Nuclear Medicine, Henri Mondor Hospital, 94000 Creteil, France; julia.chalaye@aphp.fr (J.C.);; 4EC2M3-EA7375, Cancer Research Team, Faculty of Medicine, Université Paris Est Créteil, 94000 Creteil, France

**Keywords:** PET, MRI, PET-MRI, endocrine, G-NET, gastrointestinal, pancreas

## Abstract

**Simple Summary:**

Simultaneous positron emission tomography/magnetic resonance imaging (PET-MRI) combines the high sensitivity of PET with the high specificity of MRI. Here, we aimed to evaluate whether a combined PET-MRI exam may give more information than conventional exams at the time of diagnosis in a series of patients presenting with a neuroendocrine tumor in the pancreas or within the gastrointestinal tract. We also investigated if the procedure could influence survival. In 71 patients, we performed a PETMRI at the baseline and in 50 patients the follow-up was based on PET-MRI performed every 6–12 months. The results show that the hybrid exam yielded new information at the baseline and during the follow-up, but had no impact on the survival.

**Abstract:**

Introduction and aim: Simultaneous positron emission tomography/magnetic resonance imaging (PET-MRI) combines the high sensitivity of PET with the high specificity of MRI and is a tool for the assessment of gastroenteropancreatic neuroendocrine neoplasms (G-NENs). However, it remains poorly evaluated with no clear recommendations in current guidelines. Thus, we evaluated the prognostic impact of PET-MRI in G-NEN patients. Methods: From June 2017 to December 2021, 71 G-NEN patients underwent whole-body PET-MRI for staging and/or follow-up purposes. A whole-body emission scan with ^18^F-6-fluoro-L-dihydroxyphenylalanine (^18^FDOPA, *n* = 30), ^18^F-fluoro-2-deoxy-D-glucose (^18^FDG, *n* = 21), or ^68^Ga-(DOTA(0)-Phe(1)-Tyr(3))-octreotide (^68^Ga-DOTATOC, *n* = 20) with the simultaneous acquisition of a T1-Dixon sequence and diffusion-weighed imaging (DWI), followed by a dedicated step of MRI sequences with a Gadolinium contrast was performed. The patients underwent PET-MRI every 6–12 months during the follow-up period until death. Over this period, 50 patients with two or more PET-MRI were evaluated. Results: The mean age was 61 [extremes, 31–92] years. At the baseline, PET-MRI provided new information in 12 cases (17%) as compared to conventional imaging: there were more metastases in eight, an undescribed location (myocardia) in two, and an unknown primary location in two cases. G grading at the baseline influenced overall survival. During the follow-up (7–381 months, mean 194), clinical and therapy managements were influenced by PET-MRI in three (6%) patients due to new metastases findings when neither overall, nor disease-free survivals in these two subgroups (*n* = 12 vs. *n* = 59), were different. Conclusion: Our study suggests that using PET/MRI with the appropriate radiotracer improves the diagnostic performance with no benefit on survival. Further studies are warranted to evaluate the cost-effectiveness of this procedure.

## 1. Introduction

Gastroenteropancreatic neuroendocrine neoplasms (G-NENs) are rare, although their incidence is increasing [1,2]. They include well-differentiated neuroendocrine tumors (NET) and poorly differentiated neuroendocrine carcinomas (NEC) [3]. Neuroendocrine neoplasms, known also as carcinoids, affect both men and women and may now concern younger adults since their incidence is increasing. Although they express a hormonal component, this may not cause hormonal-related symptoms. However, hormonal receptors located on the cell membranes in these tumors are used as a specific target for detection and diagnosis as well as anti-tumor therapies. The initial work-up and follow-up of patients with G-NENs include endoscopic exams, especially endoscopy ultra sound (EUS). The diagnosis also involves a CT scan that highlights vascularization followed by magnetic resonance imaging (MRI) and nuclear imaging that target hormonal production and/or hormonal receptors in tumor cells. The gallium PET exploits the somatostatin receptors located on the neoplastic cells [4,5]. Positron emission tomography (PET)-MRI and PET CT are mentioned as options in the guidelines of the National Comprehensive Cancer Network (NCCN) 2021 in the evaluation and surveillance of G-NEN [3,6]. The accuracy of these exams depends on the differentiation of tumors and the capacity to express somatostatin receptors. This procedure may not detect anaplastic cells or those poorly differentiated tumors that express low numbers of receptors. 

The hybrid PET/MRI scanner allows the simultaneous acquisition of metabolic information with various PET radiotracers and tissue microenvironments with various MRI contrasts. Its introduction in France since 2015 has progressively influenced our clinical practice and machines are now available in rare pilot centers for diagnosis and follow-up. 

PET-MRI has demonstrated significant benefits, in clinical applications in various diseases such as lympjhoma [7] or in the management of epileptic patients [8]. In lymphoma, this hybrid imaging contributes to the correct classification of the disease at initial staging which is pivotal for the therapy protocol choice [9]. In the field of epilepsy, PET-MRI may allow detecting eloquent brain areas as an epileptogenic focus, for example, the number of times sedation/anesthesia and radiation can be reduced in children before assessing medically intractable epilepsy [8]. However, the place of PET-MRI in the management of G-NEN has not been formerly evaluated so far and this exam is not yet mentioned in the French guidelines [9]. Here, we report the results of a retro-prospective study that aimed to evaluate the impact of PET-MRI and show it may be of interest by allowing the detection of both well and poorly differentiated G-NEN. As the evolution course of these tumors is long and several therapies are now available, we followed up patients using conventional imaging procedures and PET-MRI and evaluated clinical management and survival in reference to the tumor classification.

## 2. Methods

### 2.1. Population and Study Design

We performed a retrospective observational study conducted in a single academic hospital (Hôpital Henri Mondor, APHP, Créteil, France). We included consecutive patients with diagnosis of G-NEN who had at least one PET-MRI. They were enrolled between June 2017 and December 2021 at academic Henri Mondor hospital. Physicians in the center were invited to perform a PET-MRI systematically instead of PET-CT during the study period. In 2015, this evolution was recognized as real progress and considered fitting with national ethical committee recommendations. All patients included in the present study were informed about the shifting from CT to PET-MRI; they were invited to undergo PET-MRI unless they preferred PET-CT ± MRI in two separate procedures. Their consent was retained as the absence of opposition in accordance with the ethical principles and French regulatory agencies requirements (French law art.16-1 et 16-6 du Code civil: for using results and procedures data, any one patient could express her/his opposition) [10]. (The certification to collect information was obtained on 21 July 2008 under 13308133v O CNIL.) Patients were followed up in either Gastroenterology, Oncology, or Endocrinology departments of Henri Mondor hospital by using PET-MRI every 6 to 12 months; for the present study, the follow-up period extended from 7 up to 381 months (mean 194). 

The diagnosis of G-NEN was made based on clinical, biological, morphological, and histological features. The G-NEN were graded using the current WHO classification (2019) as well as differentiated G1 or G2 with low proliferation rate, as assessed by Ki67 staining on tumor slides and G3 in poorly or undifferentiated tumors showing unusually highest rates of Ki67 [11]. The hybrid PET-MRI imaging was compared to conventional (CT, MRI, and EUS) exams.

### 2.2. PET-MRI Acquisition Protocol and Data Collection

We used a Biograph mMR scanner (Siemens Healthineers, Erlangen, Germany). The tracers were either ^18^F-6-fluoro-L-dihydroxyphenylalanine (^18^FDOPA) or ^68^Ga-(DOTA(0)-Phe(1)-Tyr(3))-octreotide (^68^Ga-DOTATOC) for well-differentiated tumors and ^18^F-fluoro-2-deoxy-D-glucose (^18^FDG) for poorly differentiated tumors when available. Routinely poorly differentiated tumors display a higher proliferation rate as assessed by Ki67 and lower expressions of somatostatin receptor or functional pathway of dopamine synthesis, making them less detectable by FDOPA or DOTATOC tracers. We used simultaneous acquisition of a T1-Dixon sequence and diffusion-weighed imaging (DWI), followed by a dedicated step with various MRI sequences including, whenever needed, gadolinium (Ga) contrast enhancement. All exams were interpreted both by a nuclear imaging specialist and a radiologist; then a unique report was delivered for discussion and therapy management to multidisciplinary meeting through which decision regarding the therapy are taken. Two experts participated routinely in such multidisciplinary meetings to validate the impact of PET-MRI when compared to conventional imaging exams performed within 6 months before they could influence the care and/or if new information was provided. We used RECIST Criteria to assess progression [12]. During follow-up every 6-to-12 months, PET-MRI was scheduled and images were compared to the baseline PET-MRI.

In three cases where biopsy specimens failed to establish a clear proliferation index, we speculated on histological grading based on morphologic and accuracy of metabolic criteria regarding the tracer FDG and differentiation: FDG negative and well-differentiated were classified G1 and FDG-positive G2; Ki67 was available in all G3 tumor cases.

### 2.3. Endpoints and Statistical Analysis

The primary endpoint was the overall survival, as calculated from the baseline (histologically diagnosed) and the last visit registered in our database. We excluded deaths not related to G-NEN from the survival analysis. Quantitative variables are reported as averages (±standard deviation) and qualitative variables are reported as proportions (percentages).

We compared different group variables using chi2 or Student’s t test when appropriate, with a significant established error risk < 5% and 95% CI and Bonneferoni correction when accurate.

Survival analyses were performed with the Kaplan–Meier method. Survivals were compared between groups using Log-rank test with a *p* value set at 0.05 as significant. 

## 3. Results

### 3.1. Study Participants

Over the study period, 71 patients with G-NEN (Table 1) were enrolled for undergoing PET-MRI: 36 were men and 35 were women. The average age was 61 (range, 31–92) years. The primary location was the pancreas (*n* = 42) and the small intestine (*n* = 20); others (esophagus, stomach, colon, and rectum) were observed in seven cases and double location (pancreas and small intestine) in two patients. Full histological parameters were available in 96% of cases (*n* = 68), which allowed assigning cases as 39 (55%) G1, 23 (32%) G2, 3 (4%) G3, and 3 (4%) carcinomas (NEC). The disease was metastatic in 70% of cases with a majority (*n* = 43) of liver metastases (LM). 

Secretory functional syndromes were identified in 10 cases: 5 carcinoid, 3 gastrinoma, and 2 insulinoma syndromes. All except nine patients were considered presenting with a sporadic G-NEN. Nine (13%) patients were presented with a constitutional gene mutation: four multiple endocrine neoplasia of type 1 (MEN1) and five neurofiromatosis of type 1 (NF1). These patients (MEN 1/NF1) were either G1 (*n* = 6) or G2 (*n* = 3).

Additional concomitant neoplasia was noted in 19 out of 71 patients: 5 with prostate adenocarcinoma (without neuroendocrine differentiation), 2 with hepatocellular carcinoma (HCC), and 11 other cancers such as breast, thyroid, kidney, bladder, and lymphoma. The double primary neoplasia set the issue of the determination of the cause of the tumor progression during follow-up in two patients who had both G-NEN with liver metastases and HCC. A histology analysis was not possible for all the liver lesions. Hence, treatments were adapted to target both tumors as far as possible.

Overall, 2 patients underwent endoscopic resection (stomach and rectum), 33 had surgery with R0 resection in 18 cases, 49 had medical treatment only with 45 of somatostatin analogues, 22 had targeted therapy, and 17 had chemotherapy using 5-FU, (Cis- or Oxali-) Platin, anti VEGF-R. Additional procedures such as percutaneous ablation in the liver and trans-arterial chemoembolization (TACE) were performed in one and seven patients with metastasis, respectively.

### 3.2. Impact of PET-MRI at Baseline

At the baseline, 56% of patients (*n* = 40) had the appropriate radiotracer according to the primary location and the histology grading: with FDOPA in 30, FDG in 21, and DOTATOC in 20 cases according to G-NEN grading and the availability of the tracer in our center. Briefly, FDOP was used for intestinal G1 and G2 tumors, DOTATOC for extra intestinal G1and G2 tumors, and FDG for poorly differentiated and/or G3 tumors. There was no significant difference on the impact of PET-MRI depending on the tracer, primary localization, histology grading, or gene mutation (Yes, No), (Figure 1).

The hybrid PET-MRI imaging was compared to conventional (CT, MRI, EUS) exams, which were performed within 6 previous months with an average (SD) duration between the compared imaging and PET-MRI of 124 (95) days. 

PET-MRI provided new information in 12 cases out of 71 (17%) as compared to conventional previous imaging allowing the detection of the primary location in 2 cases (Figure 2A), the detection of a higher numbers of metastasis in 8 cases, and not previously described myocardial localization in 2 cases (Figure 2B). This gain in the tumor check-up was made with PET in three cases, with the MRI in four cases, and with both modalities in five patients. It induced the modification of therapy in five patients: surgery indication in one case, the introduction of a somatostatin analogue or chemotherapy in two cases, and the giving up of the surgery or percutaneous radiofrequency for liver metastases in the remaining.

### 3.3. Impact of PET-MRI during the Follow-Up

During follow-up period, 72% of the patients (*n* = 50) had at least one PET-MRI. The tracer was estimated to be appropriate in 88% of cases (*n* = 44). The average follow-up was 80 months ± 71. Overall, the complete response with no relapse after surgery was observed in 11 patients when 29 showed a stable disease, 13 had a progression, and a doubt about relapse and/or progression persisted in 6 patients.

PET-MRI influenced the management in three patients (1.5%) during the follow-up period. This consisted of the detection of new lesions in two cases (a myocardial location due to both PET and MRI and additional metastasis in one case due to PET). The therapy was influenced by PET-MRI that consisted of surgery in one case and the administration of chemotherapy in the remaining cases.

Twelve (17%) deaths occurred during the follow-up period, with four that were considered unrelated to G-NEN. 

### 3.4. Overall Survival and Factors Associated with Overall Survival

The 1 year, 5 year, and 10 year mean (average) overall survival rates from the diagnosis of G-NEN (Figure 3A) were estimated at 95.6% (CI 95 (80.8–100)), 89.5% (CI 95 (81.7–98.1)), and 76.7% (CI 95 (64.2–91.7)), respectively. There were no significant differences between patients with pancreatic NEN and patients with small intestinal regarding the overall survival. OS at 1 year were (95.1% (CI 95 (88.7–100)) and 94.7% (CI 95 (85.2–100), respectively), at 5 year were (89.2% (CI 95 (79.6–99.9)) and 88.4% (CI 95 (74.5–100), respectively), and at 10 year were (72.7% (CI 95 (56.2–94.1)) and 78.6% (CI 95 (58.9–100), respectively). (Figure 3B, *p* = 0.98.)

However, the survival was significantly different depending on the histology grading, which was estimated at 10 years: 94.4% (CI 95 (84.4–100)) for G1, 58.6% (CI 95 (32.3–100)) for G2, and 33.3% (CI 95 (10.8–100)) for G3. (*p* < 0.0001) (Figure 3C), when there was not a significant difference in patients with versus those without an impact on clinical management provided by the PET-MRI (*p* = 0.69, Figure 3D) at the baseline. 

All nine patients presenting with a constitutional gene mutation (e.g., MEN1 and NF1) were alieved at the 10 year follow-up period, showing a good outcome, while the OS was not different (*p* = 0.75) in 19 patients with concomitant neoplasia as compared to 52 without.

## 4. Discussion

We report here a large monocentric series of 71 patients with G-NEN who underwent PET-MRI. We established that despite its enhanced analysis of lesions, providing, in many cases, new and relevant findings, it did not significantly influence the disease course. Whether PET-MRI provided new findings regarding G-NEN characteristics or not, it did not lead to an improvement in the overall survival. Of note, the combined procedure was particularly interested in detecting myocardial localization.

We show that PET-MRI can influence findings and/or management in less than 20% and 10% of cases, respectively. However, it did not significantly modify the overall survival. In addition, systematically following patients with 6–12 month PET-MRI has a minor influence with no modification in the treatment protocol, although the relevance of them could be discussed in multidisciplinary medical meetings.

PET-MRI is a time-saving procedure compared to each exam (PET and MRI) carried out separately and has a low dose of ionizing radiation. While it is certainly a promising tool in the evaluation of G-NEN, its cost-effectiveness needs to be evaluated.

This present case series is representative of patients with G-NEN. Indeed, the baseline characteristics of the study participants were similar to those reported in the literature, with the prevalence of the pancreas and small intestine localization, and the prevalence of G1 tumors [1,3,5,9,13]. Moreover, the G grade was significantly associated with an overall better survival, as previously reported [1,14].

The studies comparing PET-MRI and PET-CT have demonstrated that PET-MRI is superior in the detection and characterization of liver lesions [15,16], i.e., 99% accuracy for PET-MRI as compared to 92% accuracy for PET-CT. Several studies demonstrated the benefit of PET-MRI in the detection of the primary tumor [17,18], with a better contrast resolution of soft tissue compared to PET-CT [19]. In our study, it allowed the detection of the primary tumor in two cases localized in the small intestine and not detected by any of the conventional imaging techniques. Beiderwellen et al. suggested that PET-MRI has a better characterization in abdominal lesions, whereas PET-CT had better performances for lung and bone metastases [20]. No neuroendocrine pulmonary tumors were included in our study and we did not compare PET-CT to PET-MRI; however, PET may be considered as more appropriate when imaging procedures aim at detecting lung and bone lesions. 

In our study, most patients with syndromic G-NEN (MEN 1/NF1) had a G1 status. No death occurred among them which reflects a better prognosis, as reported in the literature data [21].

Consequently, because patients with G-NEN have a long survival, imaging techniques without irradiation should be preferred. Therefore, PET-MRI may be an optimal option for G-NEN follow-up even though it did not influence the overall survival. 

During the follow-up, we reported a risk of developing a second cancer, particularly for urinary tract cancers [22]. Among the 19 patients with a second cancer, prostate cancer was prevalent with no neuroendocrine component [23] and no interference with the G-NEN treatment. However, HCC association, was a more challenging medical decision for therapy. It was indeed difficult to differentiate the progression of HCC-associated LM and the progression of LM secondary to G-NEN. When this discrimination was impossible, treatment options were decided to be effective for both cancers. Such a case was present in our cohort in two patients with histology evidence of concomitant HCC and G-NEN. During follow-up, we could not assess which tumor was in progression despite radiotracer uptake. A percutaneous treatment was administered to target both tumors. Nevertheless, none of these patients showed evidence of an associated germline mutation (i.e. Multiple Endocrine Neoplasia—MEN—or Neurofibromatosis NF1) that may concern 20% of G-NEN patients. Although many concomitant tumors in the present series were of poor prognosis, using PET-MRI did not influence the overall survival.

Our study is inherently limited by its retro-prospective design and by the relative heterogeneity of the studied population. However, it reflects how we managed consecutive patients despite heterogeneity in clinics. Additional limitations include its monocentric nature due to the unavailability of PET-MRI in other centers through Ile de France. A small number of patients did not get the appropriate tracer at the baseline due to the lack of supply. The latter limitation could underestimate the impact of PET-MRI. Only 2 out of 12 patients in whom PET-MRI has influenced staging at the baseline had an inappropriate tracer. The primary location was not visualized in all cases despite the hybrid exam. The histology grading of tumors was presumed in the morphology and evolution arguments in six patients because of the insufficient histology material for the Ki67 measurement. This situation is common in clinics and such extrapolation is routinely adopted. In addition, the period between PET-MRI and conventional comparative imaging could seem long with a maximum duration of 178 days (a 6 months’ time period in the study). This period seemed acceptable because of very low progression in G-NEN, although we could not rule out that the difference between the two exams in a few patients could reflect a progression rather than the real influence of the PET-MRI. Finally, no impact on the management and therapy during the follow-up period may be due to the small size of the present series.

## 5. Conclusions

PET-MRI may serve as a powerful tool in the diagnostic and therapy monitoring of G-NEN patients. Further prospective studies with higher numbers of patients and medico-economic screening are needed to evaluate whether PET-MRI can substantially modify our practice in the management of G-NEN.

## Figures and Tables

**Figure 1 cancers-16-02372-f001:**
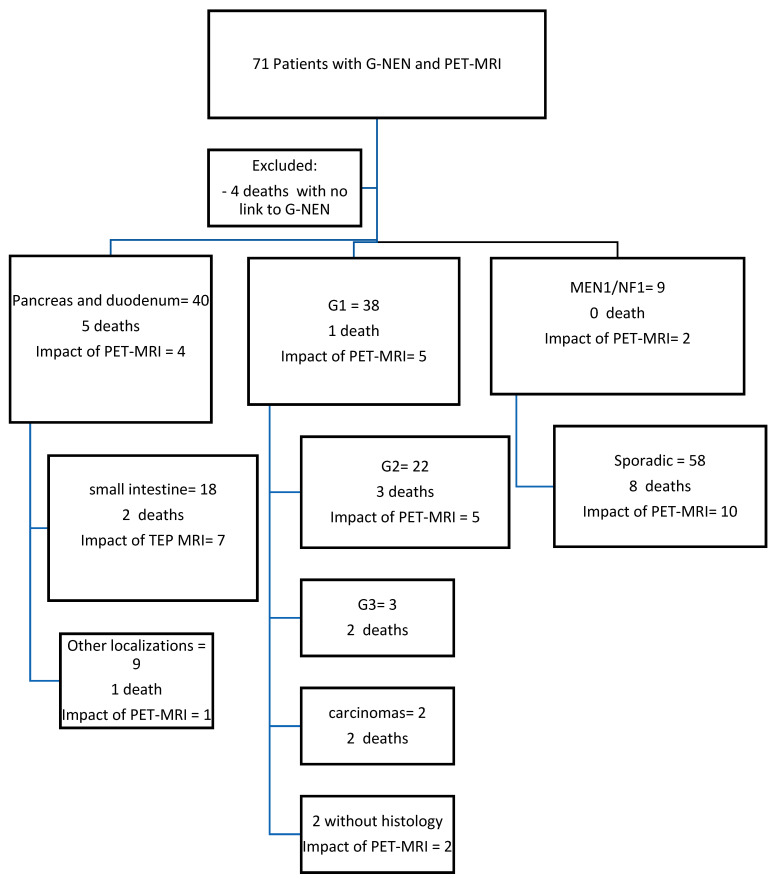
Study flow diagram. MEN1, multiple endocrine neoplasia of type 1; NF1, neurofibromatosis of type 1. Sporadic designs any G-NEN except those associated with a germline mutation (i.e. Multiple endocrine neoplasia—MEN—or Neurofibromatosis NF1).

**Figure 2 cancers-16-02372-f002:**
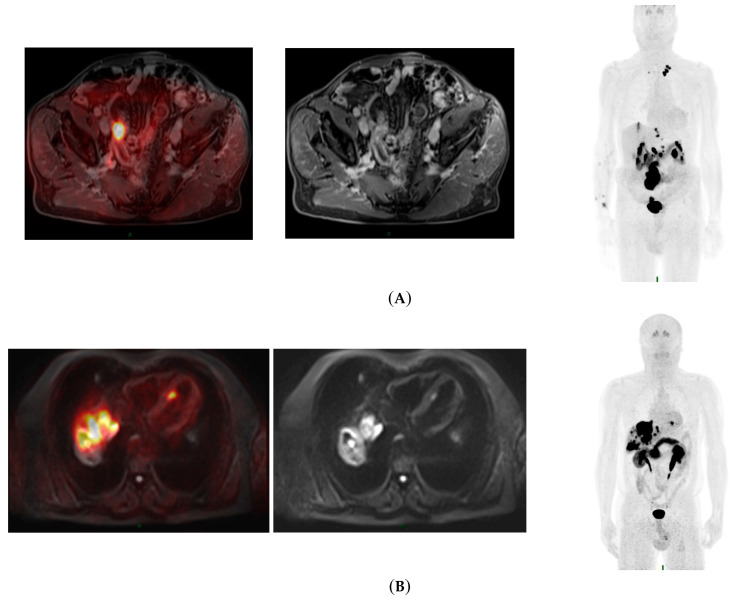
(**A**) Intense 18F-FDOPA avidity revealing primary ileal tumor in metastatic NET with previously unknown primary site. (**B**) Myocardial metastasis of NET. Intense 18F-FDOPA avidity (middle) and hypersignal in diffusion image (right).

**Figure 3 cancers-16-02372-f003:**
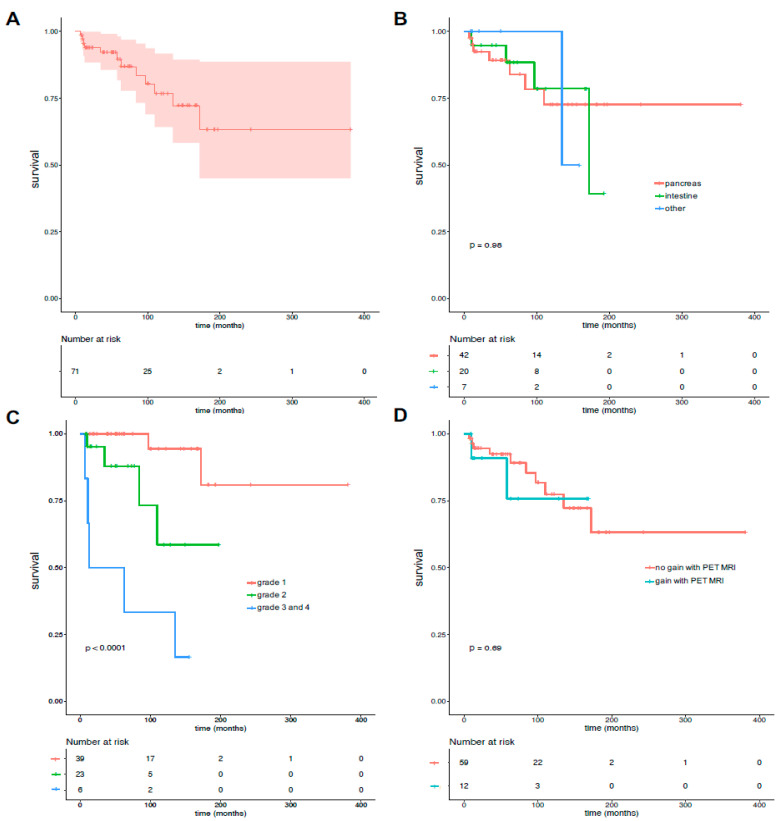
Overall survival. Overall survival (red area represents 95% confidence interval) (**A**). Survival according to G-NEN location (**B**), histologic grading (**C**), and impact of PET/MRI designed as gain or not gain at baseline (**D**). Survival was compared between groups using log-rank test, *p* value is as depicted on figures per.

**Table 1 cancers-16-02372-t001:** Characteristics of participants at baseline.

Characteristics	N = 71
Age, years (Extremes)	61 (31–92)
Female gender, *n* (%)	35 (49.3)
Primary location, *n* (%)	
Pancreas	42 (59.15)
Small intestine	20 (28.16)
Double location	2 (2.81)
Others	7 (9.85)
Metastasis, *n* (%)	
Yes	50 (70)
No	21 (30)
Histology grade, *n* (%)	
G1	39 (54.92)
G2	23 (32.39)
G3	3 (4.22)
NEC	3 (4.22)
unknown	3 (4.22)
Functional syndrome, *n* (%)	
Yes	10 (14.08)
- Carcinoid syndrome	5 (7.04)
- Gastrinoma	3 (4.22)
- Insulinoma	2 (2.81)
No	61 (85.91)
Genetic syndrome association, *n* (%)	
Yes	9 (12.67)
- MEN1	6 (8.45)
- NF1	5 (7.04)
No	62 (87.32)

NEC, neuroendocrine carcinoma; MEN1, multiple endocrine neoplasia of type 1 and NF1, neurofibromatosis of type 1 are two well-characterized germline mutations associated with G-NEN.

## Data Availability

All data supporting reported results can be found in the Table A1 in Appendix A; any additional details/data should be asked for from the corresponding author.

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
