# Peer review of "Interest of Integrated Whole-Body PET/MR Imaging in Gastroenteropancreatic Neuroendocrine Neoplasms: A Retro-Prospective Study"

_cancers, 2024, doi:10.3390/cancers16132372_

Round 1
Reviewer 1 Report
Comments and Suggestions for Authors
The researchers propose the prospective study of 71 patients affected by neuroendocrine neoplasia and studied with PTE-MRI by scanning the whole body with 18F-6-fluoro-L-dihydroxyphenylalanine (18FDOPA, n=30), 18F-fluoro- 2-deoxy-D -glucose (18FDG, n=21), or 68Ga-(DOTA(0)-Phe(1)-Tyr(3))-octreotide (68Ga-DOTATOC, n=20) with T1-Dixon sequence simultaneous and diffusion-weighted imaging (DWI). It seems appropriate to make some clarifications on the topic after having carefully read the paper. Neuroendocrine neoplasms, known until not long ago as carcinoids, a term that has not completely disappeared, are much less rare than they were about 15 years ago. They affect both men and women and while previously they appeared mainly after the age of 60, now we also see them at younger ages. They are often occasional findings given that although they express a hormonal component, this is not capable of causing symptoms as it is not a perfectly configured product. So this has no effect because it is unable to attach to the receptors located on the cell membranes. The diagnosis involves a CT scan with double contrast which highlights this lesion well, marking mainly the surface, less the central part. This is followed by gallium PET which exploits the somatostatin receptors located on the neoplastic cells. This test may not be reliable if we are faced with a NEN because it is composed of more anaplastic cells, therefore with fewer receptors. Our colleagues used an octreotide tracer for PET-MRI for the same reason. Once the diagnostic process is completed, the surgical therapeutic process follows, in the absence of secondary localizations or complications (bleeding or occlusion in the case of hollow viscera). Medical treatment involving somatostatin or analogues, which can cause complications (PMID: 38051513 if you want you can cite it). Everolimus which among other things inhibits calcineurin, therefore useful in patients with renal failure or immunotherapy. These patients will be followed up with instrumental tests, among which PET-MRI with the reported tracer certainly has an important role. This work has its own validity which in my opinion must be integrated taking into consideration the notes reported and which are certainly in the researchers' database. English to review good bibliography
Comments on the Quality of English LanguageEnglish needs to be revised
Author Response
We thank the rveiwer for his kind suggestion for adding more details in terms of the strategy and medical or surgical proceudres; we now modified our mansucript and added part of suggestion as recommended and colourd these changes
The English has been reviewed by an mother language readres and modified accordingly
Reviewer 2 Report
Comments and Suggestions for Authors
The paper is generally easy to read in good medical English but several odd word uses and several “frenchisms” must be corrected. The paper’s verbiage can be shortened without loss of information. This paper gives us a good background [but that requires some improvements] on PET-MRI and what this imaging can do and its limitations in caring for G-NETs. After corrections, it will be helpful to clinicians interested in caring for G-NETs.
A personal opinion here - no need to first spell out CT or MRI. I think they have reached the status of DNA, EKG, WHO, USA, EU, etc. PET might be best to spell out for now.
Line 36, in any case after you specified what PET, CT, and PET-MRI stood for, dont then spell out what PET-CT stands for.
Line 23. Error “. Amongst, were followed up using PET-MRI every 6-12 months.” ? Do you mean the 71 G-NET cohort ? Amongst what are you referring ?
Line 38, do not abbreviate National Comprehensive Cancer Network . you only use the abbreviation once. Limiting abbreviations makes things easier for your readers.
Line 28 Error. “benefice on survival.” should read “benefit”.
Please reword. Unclear as written. “Results: At baseline, 50 (70%) patients were metastatic with PET-MRI having influenced the clinical management in 12 (16%) patients when disease-free survivals (n= 12 vs n=59), were not different. During the follow up (7-381 months, mean 194), clinical and therapy managements were influenced by PET-MRI in 3 (6%) patients due to new metastases findings. “ .
We must know the decision process, to act or not act on any new PET-MRI findings.
If we would disagree with your decision algorithm to act or not, then we would also not agree with your conclusion that PET-MRI did not influence. It would add greatly to your paper if you devoted a section at line 42, 43, outlining how your introduction of PET-MRI has “progressively influenced our clinical practices. “ Adding this need not be lengthy but will help clinicians without actress yet to this machine understand its use better.
Again line 44 “PET-MRI has demonstrated significant benefits, in clinical applications in various diseases such as lympjhoma or in the management of epileptic patients” I think stating this then requires a few sentences at least of how.
Line 67, since your paper is aimed at general clinicians, it might be best to spell out with first use 18FDOPA or 68Ga-DOTATOC as you did in the Abstract. Also explain why “for well-differentiated tumors and 18FDG for poorly differentiated tumors when available.” At least several sentences here are required to fulfill the article’s goal didactic function.
Line 70, there is no justification for using upper case in a sentence using gadolinium. Yes, use abbreviation Gd if you wish but if you spell it out, it must be gadolinium. It is not a proper noun in French or English, the same rule applies in both.
Line 70, error. “to such multidisciplinary…” should read in such.
Line 77 error. “During follow-up, every 6 to 12 PET-MRI was scheduled…” ? every 6 to 12 what ?
Line 82. Error. You have not yet specified your grading system for G-NETs. May I suggest a little table listing the core criteria for G1, G2, and G3 ?
Line 109. “Additional concomitant neoplasia was noted in 18 out of 71 patients: 5 with prostate cancer and 2 with hepatocellular carcinoma (HCC) and 11 other cancers such as breast, thyroid, kidney, bladder, lymphoma. “ This might be the most interesting finding of your paper. How do we know the prostate cancer was not a primary prostate adenocarcinoma with neuroendocrine differentiation ? A mixed single tumor ? Similarly to the others. Given that 25% of your PET-MRIs disclosed another malignancy, I dont think that you can conclude “Using PET/ MRI with the appropriate radiotracer improves diagnostic performance with no benefice on survival. “ Presumably the non-G-NET tumors you uncovered were treated also and did that not improve these people’s survival ? This seems to be a major finding, the major finding of your work. Doesn’t it ?
Line 124-127 Again here the need to explain what the differences in the three radiotracers is crucial for readers to understand your work. You are addressing general physicians, not nuclear medicine physicians.
Re. Figure 1, I don’t understand what the box “sporadic” means.
Line 135, seems wrong. “PET-MRI provided new information in 12 cases out of 71 (17%) as compared to conventional previous imaging allowing detection of the primary “. Didn’t the finding of 25% of your cohort had another, potentially more serious and more treatable malignancy count as important new information ?
Re. Figure 3, Before accepting this paper I would require adding a line including data on the 25% who had another cancer diagnosed on all 4 graphs.
Line 207, “more performant for the detection…” is odd and vague but readers might be able to guess what you intend. Can you rephrase ? Performant for what, on what measure ?
Re line 225, “Our study is inherently limited by its retro-prospective design and by the relative heterogeneity of the studied population.” maybe not. Like many limitations this heterogeneity is also a plus. A strong one in my opinion.
Comments on the Quality of English Language
The paper is generally easy to read in good medical English but several odd word uses and several “frenchisms” must be corrected. The paper’s verbiage can be shortened without loss of information. This paper gives us a good background [but that requires some improvements] on PET-MRI and what this imaging can do and its limitations in caring for G-NETs. After corrections, it will be helpful to clinicians interested in caring for G-NETs.
A personal opinion here - no need to first spell out CT or MRI. I think they have reached the status of DNA, EKG, WHO, USA, EU, etc. PET might be best to spell out for now.
Line 36, in any case after you specified what PET, CT, and PET-MRI stood for, dont then spell out what PET-CT stands for.
Line 23. Error “. Amongst, were followed up using PET-MRI every 6-12 months.” ? Do you mean the 71 G-NET cohort ? Amongst what are you referring ?
Line 38, do not abbreviate National Comprehensive Cancer Network . you only use the abbreviation once. Limiting abbreviations makes things easier for your readers.
Line 28 Error. “benefice on survival.” should read “benefit”.
Please reword. Unclear as written. “Results: At baseline, 50 (70%) patients were metastatic with PET-MRI having influenced the clinical management in 12 (16%) patients when disease-free survivals (n= 12 vs n=59), were not different. During the follow up (7-381 months, mean 194), clinical and therapy managements were influenced by PET-MRI in 3 (6%) patients due to new metastases findings. “ .
We must know the decision process, to act or not act on any new PET-MRI findings.
If we would disagree with your decision algorithm to act or not, then we would also not agree with your conclusion that PET-MRI did not influence. It would add greatly to your paper if you devoted a section at line 42, 43, outlining how your introduction of PET-MRI has “progressively influenced our clinical practices. “ Adding this need not be lengthy but will help clinicians without actress yet to this machine understand its use better.
Again line 44 “PET-MRI has demonstrated significant benefits, in clinical applications in various diseases such as lympjhoma or in the management of epileptic patients” I think stating this then requires a few sentences at least of how.
Line 67, since your paper is aimed at general clinicians, it might be best to spell out with first use 18FDOPA or 68Ga-DOTATOC as you did in the Abstract. Also explain why “for well-differentiated tumors and 18FDG for poorly differentiated tumors when available.” At least several sentences here are required to fulfill the article’s goal didactic function.
Line 70, there is no justification for using upper case in a sentence using gadolinium. Yes, use abbreviation Gd if you wish but if you spell it out, it must be gadolinium. It is not a proper noun in French or English, the same rule applies in both.
Line 70, error. “to such multidisciplinary…” should read in such.
Line 77 error. “During follow-up, every 6 to 12 PET-MRI was scheduled…” ? every 6 to 12 what ?
Line 82. Error. You have not yet specified your grading system for G-NETs. May I suggest a little table listing the core criteria for G1, G2, and G3 ?
Line 109. “Additional concomitant neoplasia was noted in 18 out of 71 patients: 5 with prostate cancer and 2 with hepatocellular carcinoma (HCC) and 11 other cancers such as breast, thyroid, kidney, bladder, lymphoma. “ This might be the most interesting finding of your paper. How do we know the prostate cancer was not a primary prostate adenocarcinoma with neuroendocrine differentiation ? A mixed single tumor ? Similarly to the others. Given that 25% of your PET-MRIs disclosed another malignancy, I dont think that you can conclude “Using PET/ MRI with the appropriate radiotracer improves diagnostic performance with no benefice on survival. “ Presumably the non-G-NET tumors you uncovered were treated also and did that not improve these people’s survival ? This seems to be a major finding, the major finding of your work. Doesn’t it ?
Line 124-127 Again here the need to explain what the differences in the three radiotracers is crucial for readers to understand your work. You are addressing general physicians, not nuclear medicine physicians.
Re. Figure 1, I don’t understand what the box “sporadic” means.
Line 135, seems wrong. “PET-MRI provided new information in 12 cases out of 71 (17%) as compared to conventional previous imaging allowing detection of the primary “. Didn’t the finding of 25% of your cohort had another, potentially more serious and more treatable malignancy count as important new information ?
Re. Figure 3, Before accepting this paper I would require adding a line including data on the 25% who had another cancer diagnosed on all 4 graphs.
Line 207, “more performant for the detection…” is odd and vague but readers might be able to guess what you intend. Can you rephrase ? Performant for what, on what measure ?
Re line 225, “Our study is inherently limited by its retro-prospective design and by the relative heterogeneity of the studied population.” maybe not. Like many limitations this heterogeneity is also a plus. A strong one in my opinion.
Author Response
Response to reviewer 2
Line 36, in any case after you specified what PET, CT, and PET-MRI stood for, dont then spell out what PET-CT stands for.
Done
Line 23. Error “. Amongst, were followed up using PET-MRI every 6-12 months.” ? Do you mean the 71 G-NET cohort ? Amongst what are you referring ? 50 out of 71
Line 38, do not abbreviate National Comprehensive Cancer Network . you only use the abbreviation once. Limiting abbreviations makes things easier for your readers. Done
Line 28 Error. “benefice on survival.” should read “benefit”. Done
Please reword. Unclear as written. “Results: At baseline, 50 (70%) patients were metastatic with PET-MRI having influenced the clinical management in 12 (16%) patients when disease-free survivals (n= 12 vs n=59), were not different. During the follow up (7-381 months, mean 194), clinical and therapy managements were influenced by PET-MRI in 3 (6%) patients due to new metastases findings. “ .
This is now re phrases
We must know the decision process, to act or not act on any new PET-MRI findings.
If we would disagree with your decision algorithm to act or not, then we would also not agree with your conclusion that PET-MRI did not influence. It would add greatly to your paper if you devoted a section at line 42, 43, outlining how your introduction of PET-MRI has “progressively influenced our clinical practices. “ Adding this need not be lengthy but will help clinicians without actress yet to this machine understand its use better.
We explained in the results & discussion how this impacted our decision in each patient including those with concomitant neoplasia; however there was not a significant impact of PET-MRI on the decision care and OS was not impacted
Again line 44 “PET-MRI has demonstrated significant benefits, in clinical applications in various diseases such as lympjhoma or in the management of epileptic patients” I think stating this then requires a few sentences at least of how.
There are now more details in the introduction section
Line 67, since your paper is aimed at general clinicians, it might be best to spell out with first use 18FDOPA or 68Ga-DOTATOC as you did in the Abstract. Also explain why “for well-differentiated tumors and 18FDG for poorly differentiated tumors when available.” At least several sentences here are required to fulfill the article’s goal didactic function.
We now detailed in introduction and in methods
Line 70, there is no justification for using upper case in a sentence using gadolinium. Yes, use abbreviation Gd if you wish but if you spell it out, it must be gadolinium. It is not a proper noun in French or English, the same rule applies in both.
Correct, done
Line 70, error. “to such multidisciplinary…” should read in such.
Done
Line 77 error. “During follow-up, every 6 to 12 PET-MRI was scheduled…” ? every 6 to 12 what ?
Done
Line 82. Error. You have not yet specified your grading system for G-NETs. May I suggest a little table listing the core criteria for G1, G2, and G3 ?
More detailed in introduction and methods sections
Line 109. “Additional concomitant neoplasia was noted in 18 out of 71 patients: 5 with prostate cancer and 2 with hepatocellular carcinoma (HCC) and 11 other cancers such as breast, thyroid, kidney, bladder, lymphoma. “ This might be the most interesting finding of your paper. How do we know the prostate cancer was not a primary prostate adenocarcinoma with neuroendocrine differentiation ? A mixed single tumor ? Similarly to the others. Given that 25% of your PET-MRIs disclosed another malignancy, I dont think that you can conclude “Using PET/ MRI with the appropriate radiotracer improves diagnostic performance with no benefice on survival. “ Presumably the non-G-NET tumors you uncovered were treated also and did that not improve these people’s survival ? This seems to be a major finding, the major finding of your work. Doesn’t it ?
Indeed; the present review indicates 19/71 patients have been presenting with another neoplastic disease (one more breast cancer), however the overall survival didn’t change by performing PET-IRM every 6-12 months during the follow up we now included details in the results and discussion and give a figure just for the reviewer indication that survival curves are exactly similar
Line 124-127 Again here the need to explain what the differences in the three radiotracers is crucial for readers to understand your work. You are addressing general physicians, not nuclear medicine physicians.
This is now mentioned in methods and results sections
Re. Figure 1, I don’t understand what the box “sporadic” means.
Reference to Genetic disease versus sporadic is now mentioned in legends to the figure
Line 135, seems wrong. “PET-MRI provided new information in 12 cases out of 71 (17%) as compared to conventional previous imaging allowing detection of the primary “. Didn’t the finding of 25% of your cohort had another, potentially more serious and more treatable malignancy count as important new information ?
Although some of second tumours are known to have poorer outcome than the G-NEN, there was no impact on overall survival related to the second tumour
Re. Figure 3, Before accepting this paper I would require adding a line including data on the 25% who had another cancer diagnosed on all 4 graphs.
This is now explained in the discussion section
Line 207, “more performant for the detection…” is odd and vague but readers might be able to guess what you intend. Can you rephrase ? Performant for what, on what measure ?
Done
Re line 225, “Our study is inherently limited by its retro-prospective design and by the relative heterogeneity of the studied population.” maybe not. Like many limitations this heterogeneity is also a plus. A strong one in my opinion.
We added in the discussion now that this failure is balanced by routine pragmatically medical care

Round 2
Reviewer 1 Report
Comments and Suggestions for Authors
The corrections brought to the work are certainly acceptable, we must add a further evaluation, the PET-MRI with the tracer, the object of the work, can highlight other neoplasms such as paraganglioma where somatostatin receptors are present, or may not be suitable for the total absence of receptors because highly undifferentiated tumors. This will obviously also affect the therapy with the a priori exclusion of somatostatin and analogues. However, it is certainly a diagnostic to be taken into high consideration for the diagnosis and above all in the follow-up of patients suffering from this insidious and increasingly frequent disease. It is advisable to add some references
Comments on the Quality of English Languageenglish needs to be revised
Author Response
|
|
Response to Reviewer 2 Comments The corrections brought to the work are certainly acceptable, we must add a further evaluation, the PET-MRI with the tracer, the object of the work, can highlight other neoplasms such as paraganglioma where somatostatin receptors are present, or may not be suitable for the total absence of receptors because highly undifferentiated tumors. This will obviously also affect the therapy with the a priori exclusion of somatostatin and analogues. However, it is certainly a diagnostic to be taken into high consideration for the diagnosis and above all in the follow-up of patients suffering from this insidious and increasingly frequent disease. It is advisable to add some references.
Thank you very much for taking the time to review this manuscript. Please find the detailed responses below and the corresponding revisions/corrections highlighted/in track changes in the re-submitted files.
|
|||
|
Point-by-point response to Comments and Suggestions for Authors |
|
|
|
|
|
|
Comments 1: The corrections brought to the work are certainly acceptable |
|||
|
Response 1 Thank you for pointing this out.
|
|
|
|
|
|
Comments 2: PET-MRI with the tracer, the object of the work, can highlight other neoplasms such as paraganglioma where somatostatin receptors are present |
|
|
|
|
|
Response 2: Agree. Indeed, this is the aim of the study
Comments 3. may not be suitable for the total absence of receptors because highly undifferentiated tumors Response 3: Agree. This is the reason for which different tracers have been currently used
Comments 4 : However, it is certainly a diagnostic to be taken into high consideration for the diagnosis and above all in the follow-up of patients suffering from this insidious and increasingly frequent disease. It is advisable to add some references
Response 4: We do not understand what is requested here; we included all relevant references related to the present work and do not feel any new reference is needed unless the reviewer clarify what is expected here
|
|
|
|
|
